# IgM-enriched immunoglobulin improves colistin efficacy in a pneumonia model by *Pseudomonas aeruginosa*

Tania Cebrero-Cangueiro[1,2], Gema Labrador-Herrera[1,2], Marta Carretero-Ledesma[1,2], Soraya Herrera-Espejo[1,2], Rocío Álvarez-Marín[1,2,3], Jerónimo Pachón[2,3,4] (iD), José Miguel Cisneros[1,2,3], María Eugenia Pachón-Ibáñez[1,2,3] (iD)

**We evaluated the efficacy of ceftazidime or colistin in combination with polyclonal IgM-enriched immunoglobulin (IgM-IG), in an experimental pneumonia model (C57BL/6J male mice) using two multidrug-resistant *Pseudomonas aeruginosa* strains, both ceftazidime-susceptible and one colistin-resistant. Pharmacodynamically optimised antimicrobials were administered for 72 h, and intravenous IgM-IG was given as a single dose. Bacterial tissues count and the mortality were analysed. Ceftazidime was more effective than colistin for both strains. In mice infected with the colistin-susceptible strain, ceftazidime reduced the bacterial concentration in the lungs and blood ($-2.42$ and $-3.87$ $\log_{10}$ CFU/ml) compared with colistin ($-0.55$ and $-1.23$ $\log_{10}$ CFU/ml, respectively) and with the controls. Colistin plus IgM-IG reduced the bacterial lung concentrations of both colistin-susceptible and resistant strains ($-2.91$ and $-1.73$ $\log_{10}$ CFU/g, respectively) and the bacteraemia rate of the colistin-resistant strain ($-44\%$). These results suggest that IgM-IG might be useful as an adjuvant to colistin in the treatment of pneumonia caused by multidrug-resistant *P. aeruginosa*.**

## Introduction

*Pseudomonas aeruginosa* is a Gram-negative bacillus (GNB) that causes community- and healthcare-associated infections, especially in intensive care units (Bassetti et al, 2016) and in patients with chronic underlying diseases, including pneumonia, urinary tract infections, and bloodstream infections (BSIs) (Mehrad et al, 2015). This microorganism is one of the paradigms of the global threat of antimicrobial resistance (Cabot et al, 2016). Thus, physicians worldwide are facing a growing challenge from an increasing number of difficult-to-treat multidrug-resistant *P. aeruginosa* infections and consequent increases in morbidity and mortality rates (Lodise et al, 2007; Bauer et al, 2013). Therefore, *P. aeruginosa* is considered a critical pathogen for the research and development of new antibiotics (Tacconelli et al, 2018). Although in the last years several antibiotics have been approved for human use (Lupia et al, 2020), new antimicrobials cannot be produced by industry and approved for their clinical use at a rate equivalent to that in which antibiotic resistance is acquired (Poole, 2011; Pachori et al, 2019). For these reasons, new therapeutic alternatives to treat this pathogen are necessary (Witney et al, 2014; Tacconelli et al, 2018).

The modulation of the immune system, as an adjuvant to antimicrobials, has been analysed in several studies, which have suggested that combining immunoglobulins with antibiotics could improve the prognosis of human infections (Schedel et al, 1991; Werdan & Pilz, 1996; Kreymann et al, 2007; Laupland et al, 2007). IgG is the most abundant antibody and provides protection against infections, bacterial and/or viral. Among its most important functions are to neutralize and eliminate the pathogen, in this case *P. aeruginosa*, toxins generated by the bacteria, and/or other substances produced in inflammatory processes or cell destruction (Birdsall & Casadevall, 2020). Nevertheless, IgG can take some time to form after infection, so combining it with IgM, first line of defense against infection by inhibiting bacterial adhesion to epithelial cells and neutralizing toxins could increase the immune response and be an adjuvant to antibiotics (Birdsall & Casadevall, 2020). In addition, IgM combined with IgG and IgA have shown opsonophagocytic activity against carbapenem-resistant *P. aeruginosa* (Rossmann et al, 2015). These reasons suggest that using IgM-IG could modulate the immune system of the mice increasing the efficacy of the tested antibiotics. Other approaches to immunomodulation, different to immunoglobulins, have also been reported. In murine experimental models infected with ceftazidime-resistant *P. aeruginosa*, a combination of lysophosphatidylcholine—a modulator of the immune system—with ceftazidime and/or imipenem led to a greater decrease in the bacterial concentration in tissues compared with when these antibiotics were used as monotherapies (Parra-Millán et al, 2020). Modulation of the immune system might circumvent antibacterial resistance, leading to an efficacious treatment of these infections (Chiang et al, 2018), does not disrupt microbiota (Schubert et al, 2015), and protects against reinfection (Quintin et al, 2012).

---

[1]Unit of Infectious Diseases, Microbiology, and Preventive Medicine, Virgen del Rocío University Hospital, Seville, Spain    [2]Institute of Biomedicine of Seville (IBiS), Virgen del Rocío University Hospital/CSIC/ University of Seville, Seville, Spain    [3]Centro de Investigación Biomédica en Red de Enfermedades Infecciosas, Madrid, Spain    [4]Department of Medicine, University of Seville, Seville, Spain

Correspondence: mpachon-ibis@us.es

**Table 1.  MIC/MBC of different antibiotics for the two clinical *Pseudomonas aeruginosa* strains.**

| Strains | MIC/MBC (mg/l) | | | | | | | | | |
|---|---|---|---|---|---|---|---|---|---|---|
| | AMK | GEN | TIC | MEM | CAZ | FEP | SBT | CST | CIP | TGC |
| Pa147 | 1/2 | >128/128 | 32/64 | >128/>128 | 8/32 | 8/16 | >128/>128 | 0.5/4 | 32/64 | 4/8 |
| PaM1 | 4/16 | 1/2 | 1/32 | 0.25/2 | ≤0.25/2 | 2/32 | 128/>128 | >128/>128 | 1/2 | 4/8 |

MIC, minimum inhibitory concentrations; MBC, minimum bactericidal concentration; AMK, amikacin; GEN, gentamicin; TIC, ticarcillin; MEM, meropenem; CAZ, ceftazidime; FEP, cefepime; SBT, sulbactam; CST, colistin; CIP, ciprofloxacin; TGC, tigecycline. MIC results were interpreted according to EUCAST breakpoints: AMK, susceptible, MIC ≤ 8 mg/l and resistant MIC > 16 mg/l; GEN, susceptible, MIC ≤ 4 mg/l and resistant MIC > 4 mg/l; TIC, susceptible, MIC ≤ 16 mg/l and resistant MIC > 16 mg/l; MER, susceptible, MIC ≤ 2 mg/l and resistant MIC > 8 mg/l; CAZ and FEP, susceptible, MIC ≤ 8 mg/l and resistant MIC > 8 mg/l; CST, susceptible, MIC ≤ 2 mg/l and resistant MIC > 2 mg/l; CIP susceptible, MIC ≤ 0.5 mg/l and resistant MIC > 0.5 mg/l. Currently, there are no established susceptibility criteria in either EUCAST or CLSI for tigecycline and sulbactam in *P. aeruginosa*. Studies were performed in triplicate in different days.

A prospective, randomized clinical trial showed that antibiotics in combination with iv polyclonal immunoglobulin (Ig) preparation containing IgG, IgM, and IgA as an adjunctive therapy were associated with significantly improved survival in patients with septic shock (Schedel et al, 1991). Giamarellos-Bourboulis et al (2016), using a case–control design, have reported that the study group with a combination of antibiotics plus polyclonal IgM-enriched immunoglobulins (IgM-IG) presented a survival rate higher (61% versus 42%) compared with the comparator group (antibiotics only) until day 28 in patients with sepsis or septic shock caused by multidrug-resistant GNBs, including *P. aeruginosa*.

Therefore, the aim of this study was to evaluate the efficacy of ceftazidime or colistin in combination with IgM-enriched immunoglobulins (IgM-IG) in treating pneumonia in a murine model, where infection was induced using clinical strains of multidrug-resistant *P. aeruginosa*.

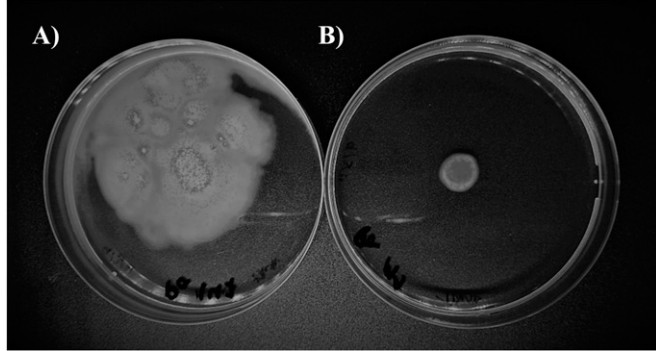

**Figure 1.  Surface motility of the two clinical *Pseudomonas aeruginosa* strains. (A, B)** Colistin-susceptible Pa147 (MIC = 0.5 mg/l) and (B) colistin-resistant PaM1 (MIC ≥ 128 mg/l). Bacterial suspension (3 μl) was plated onto LB containing 0.3% agarose. Plates were incubated during 24 h at 37°C with 80% humidity, and the radio of surface extensions was measured.

# Results

### Antimicrobial susceptibility testing

MICs and MBCs for the two clinical strains are shown in Table 1. Both strains were susceptible to ceftazidime, whereas only Pa147 was susceptible to colistin.

### *Surface motility assay*

Motility assays are shown in Fig 1. The colistin-resistant PaM1 strain had reduced surface motility (4 ± 0.05 mm) compared with the colistin-susceptible Pa147 strain (400 ± 0.06 mm) (*P* < 0.001).

### In vitro competition indices

When grown alone, both bacterial strains showed similar growth at 24 h. The colistin-resistant PaM1 strain showed a loss of fitness compared with the colistin-susceptible Pa147 strain (CI: 0.02, 0.01, 0.0, and 0.0 at 2, 4, 8, and 24 h, respectively) (Fig 2).

### Biofilm formation analysis

The colistin-susceptible Pa147 strain showed higher biofilm formation (97.1%) than the colistin-resistant PaM1 strain (41.2%) (Fig 3).

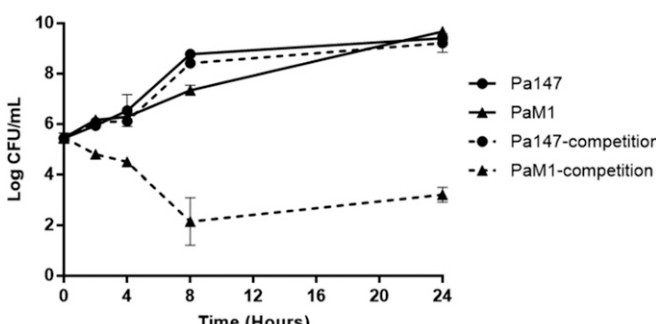

**Figure 2.  In vitro growth of colistin-susceptible Pa147 and colistin-resistant PaM1 strains in Mueller Hinton broth alone (MHB) (solid lines) and in competition (dash lines) were evaluated for 24 h.** Data are represented as means ± SD (n = 3 replicates performed in different days).

### Efficacy studies

The efficacies of colistin and ceftazidime as monotherapies and in combination with IgM-IG are shown in Table 2 and Figs 4 and 5. Compared with the control group, in the mice infected with the Pa147 strain, ceftazidime monotherapy significantly reduced the bacterial concentration in the lungs (–2.42 $\log_{10}$ CFU/g) and blood (–3.87 $\log_{10}$ CFU/ml) as well as the mortality (–40%). Colistin monotherapy

and IgM-IG administered alone only reduced the bacterial concentrations of the colistin-susceptible Pa147 strain in the lungs (−0.55 and −1.29 $\log_{10}$ CFU/g, respectively). In mice infected with the colistin-resistant PaM1 strain, ceftazidime monotherapy reduced the bacterial concentration in the lungs (−1.42 $\log_{10}$ CFU/g) and the mortality (−40%). Colistin monotherapy and IgM-IG

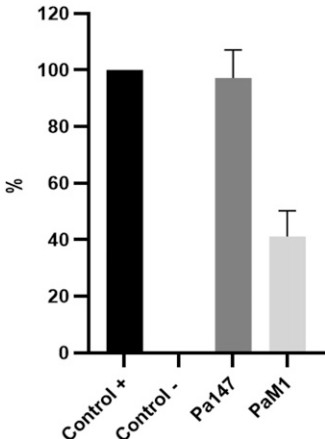

**Figure 3. Biofilm production of both *Pseudomonas aeruginosa* strains; the colistin-susceptible Pa147 and the colistin-resistant PaM1 strains.**
*Acinetobacter baumannii* ATCC 19606 and *A. baumannii* CR17 strains were used as positive and negative control, respectively. Biofilm formation was quantified by measuring the OD at 580 nm. Bars represent the mean of three separate assays performed in three different days, with error bars representing the SD.

administered alone did not have a therapeutic effect on any variables in mice infected with the PaM1 strain.

Compared with ceftazidime monotherapy, IgM-IG plus ceftazidime reduced the bacterial concentration in the blood (−0.83 $\log_{10}$ CFU/ml) and the mortality (−20%) in mice infected with the Pa147 strain, and it decreased the bacterial concentration in the lungs (−0.8 $\log_{10}$ CFU/g) and blood (−0.89 $\log_{10}$ CFU/ml) in those infected with the PaM1 strain, but the effects were not significant in any of these cases. In the secondary analysis, IgM-IG plus ceftazidime decreased bacterial concentrations in lungs (−1.86 $\log_{10}$ CFU/g, $P <$ 0.05) and blood (−3.47 $\log_{10}$ CFU/ml, $P <$ 0.05), and the mortality (−60%, $P <$ 0.05) respect to colistin monotherapy against the Pa147 strain, without differences with colistin for the PaM1 strain.

Finally, compared with colistin monotherapy, IgM-IG plus colistin significantly reduced the bacterial concentration of the colistin-susceptible Pa147 strain in the lungs (−2.36 $\log_{10}$ CFU/g). This combination also reduced, although not significantly, the bacterial concentration in the blood (−1.29 $\log_{10}$ CFU/ml) and the frequencies of bacteraemia and mortality (−30%) against the same strain. In mice infected with the colistin-resistant PaM1 strain, IgM-IG plus colistin reduced, although not significantly, the bacterial concentration in the lungs (−1.04 $\log_{10}$ CFU/g) and blood (−1.88 $\log_{10}$ CFU/ml) and the frequency of bacteraemia (−44%). In the secondary analysis IgM-IG plus colistin compared with ceftazidime monotherapy only decreased the bacteraemia rate (−44%, $P <$ 0.05) for the PaM1 strain.

The eta-square analysis showed a large size effect, in all treatment comparisons, for the bacterial concentrations in lungs (0.4863023 and 0.398475, for Pa147 and PaM1, respectively)

**Table 2. Efficacy studies of colistin and ceftazidime in monotherapy and their combinations with IgM-enriched immunoglobulins in the pneumonia murine model by *P. aeruginosa*.**

| Strains | Treatments | n | Lungs ($\log_{10}$ CFU/g) | Blood ($\log_{10}$ CFU/ml) | Bacteraemia % (n) | Mortality % (n) |
|---|---|---|---|---|---|---|
| Pa147 | CON | 10 | 9.66 ± 0.12 | 6.51 ± 0.29 | 100 (10) | 100 (10) |
| | IgM-IG | 10 | 8.37 ± 0.34a | 6.62 ± 0.28 | 100 (10) | 90 (9) |
| | CMS | 10 | 9.11 ± 0.11[a] | 5.28 ± 0.58 | 100 (10) | 100 (10) |
| | CAZ | 10 | 7.24 ± 0.38[a,c] | 2.64 ± 0.73[a,b] | 70 (7) | 60 (6)[a,c] |
| | CMS + IgM-IG | 10 | 6.75 ± 0.65[a,c] | 3.99 ± 0.89 | 70 (7) | 70 (7) |
| | CAZ + IgM-IG | 10 | 7.25 ± 0.30[a,c] | 1.81 ± 0.35[a,b,c] | 80 (8) | 40 (4)[a,c] |
| PaM1 | CON | 10 | 9.63 ± 0.12 | 6.37 ± 0.42 | 100 (10) | 100 (10) |
| | IgM-IG | 8 | 9.42 ± 0.40 | 6.78 ± 0.40 | 100 (8) | 75 (6) |
| | CMS | 10 | 8.94 ± 0.33 | 5.24 ± 0.65 | 100 (10) | 70 (7) |
| | CAZ | 10 | 8.21 ± 0.31[a,b] | 5.24 ± 0.54 | 100 (10) | 60 (6)[a] |
| | CMS + IgM-IG | 9 | 7.90 ± 0.43[a,b] | 3.36 ± 1.07 | 56 (5)[a,b,c,d,e] | 89 (8) |
| | CAZ + IgM-IG | 9 | 7.41 ± 0.40[a,b] | 4.35 ± 0.64 | 100 (9) | 67 (6)[a] |

Quantitative bacterial cultures in the lungs ($\log_{10}$ CFU/g) and blood ($\log_{10}$ CFU/ml), expressed as means ± SEM (n = 8–10 mice/for each group). CON, control; IgM-IG, IgM-enriched immunoglobulins; CMS, colistimethate sodium; CAZ, ceftazidime. $P <$ 0.05 was considered to indicate significance (analysis of variance and the Games–Howell [blood] and Tukey's [lungs] post hoc tests between means were used). Analysis of the mortality rates (%) and positive qualitative blood cultures (%), were performed using the two-tailed Fisher's test.
[a]$P <$ 0.05 respect to CON group.
[b]$P <$ 0.05 respect to IgM-IG.
[c]$P <$ 0.05 respect to CMS.
[d]$P <$ 0.05 respect to CAZ group.
[e]$P <$ 0.05 respect to CAZ + IgM-IG group.

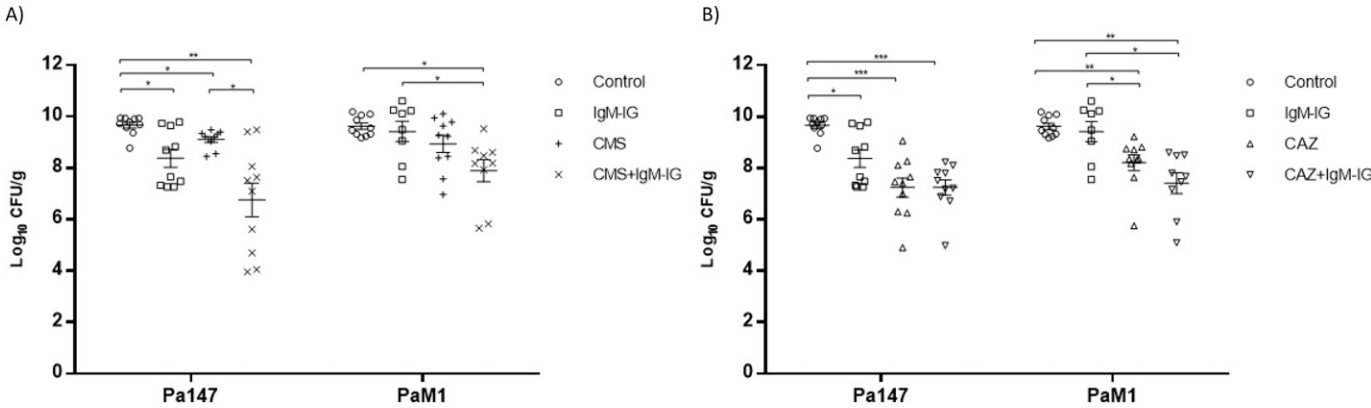

**Figure 4. In vivo efficacy of colistin or ceftazidime monotherapies and in combination with IgM-enriched inmunoglobulins on lungs bacterial concentrations. (A, B)** Efficacy of colistin (A) or ceftazidime (B) monotherapies and in combination with IgM-enriched immunoglobulins (IgM-IG) on lungs bacterial concentrations (means ± SEM) in the experimental pneumonia model by *Pseudomonas aeruginosa*. *$P < 0.05$; **$P < 0.01$; ***$P < 0.001$.

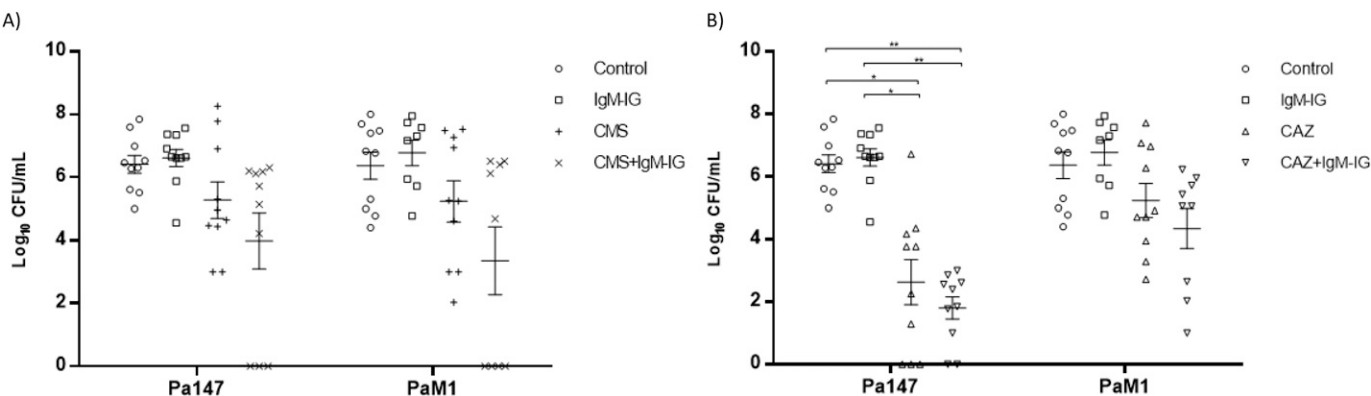

**Figure 5. In vivo efficacy of colistin or ceftazidime monotherapies and in combination with IgM-enriched inmunoglobulins on blood bacterial concentrations. (A, B)** Efficacy of colistin (A) or ceftazidime (B) monotherapies and in combination with IgM-enriched immunoglobulins (IgM-IG) on blood bacterial concentrations (means ± SEM) in the experimental pneumonia model by *Pseudomonas aeruginosa*. *$P < 0.01$; **$P < 0.001$.

and in blood (0.530371 and 0.258064, for Pa147 and PaM1, respectively).

## Discussion

Our results show that ceftazidime monotherapy was better than colistin in treating experimental murine pneumonia caused by both strains of *P. aeruginosa*, regardless of the MIC. The weak efficacy of colistin monotherapy in treating infections with the colistin-susceptible strain was significantly improved when combined with IgM-IG in terms of bacterial clearance from the lungs and blood and the bacteraemia and mortality rates. However, the combination of IgM-IG and ceftazidime led to only modest improvements compared with ceftazidime monotherapy, and the difference was not significant.

In these results, we found that the iv treatment of a single dose of IgM-IG was able to reduce the bacterial load in lungs against the colistin-susceptible strain Pa147 (−1.29 $\log_{10}$ CFU/g) when compared with the control group. IgM is the first line of defense against infection by inhibiting bacterial adhesion to epithelial cells and

neutralizing toxins (Birdsall & Casadevall, 2020). Moreover, IgM is the main immunoglobulin isotype for complement-mediated killing and complement-dependent phagocytosis of infectious bacteria because of its ability for effective complement activation (Spiegelberg, 1989). Furthermore, as is well known, the role of LPS in the *P. aeruginosa* pathogenesis, and that polysaccharides (including LPS) are T-cell–independent antigens, being antibodies induced in response to them mostly of the IgM isotype. In acute *P. aeruginosa* infection, the LPS-phenotype predominates and, as such, might be that treating the infection with antibiotics in combination with IgM-IG could be an option for acute infections because IgM-IG could awaken the immune response faster, helping to resolve it along with antibiotics. In this regard, a murine pneumonia model in BALB/c mice (Horn et al, 2010) by *P. aeruginosa* shows that the monoclonal IgM KBPA101 with exclusive reactivity with LPS of serotype O11, administered previously to the bacterial inoculation, reduced the bacterial lungs and spleen concentrations at 48 h. In other pneumonia model in C57BL/6 mice (Secher et al, 2011), the same monoclonal IgM KBPA101 could be detected in the bronchoalveolar lavage fluid of *P. aeruginosa* infected mice but not in uninfected animals, after the iv administration

and, when it was given 4 h after the inoculation, decreased (1.5 $\log_{10}$ CFU) significantly the bacterial lungs concentration. IgM KBPA101 could be detected in the bronchoalveolar lavage fluid of infected mice but not in uninfected animals. In addition, this monoclonal antibody increased the neutrophils recruitment in the lungs compared with control PBS-treated infected mice, whereas macrophage/monocyte recruitment did not (Secher et al, 2011). Although the IgM-IG is different to monoclonal IgM KBPA101, these data may explain the better results observed regarding bacterial concentrations, in lungs compared with blood in the present study.

The colistin dosage used in this study has been proven to be effective in several murine studies using clinical multidrug-resistant GNB strains susceptible to colistin (MIC = 0.5 mg/l) (Cai et al, 2018; Pachón-Ibáñez et al, 2018). Nevertheless, when used in the present study, colistin monotherapy was not effective in clearing the bacteria from the blood or in reducing positive blood cultures and mortality rates in the pneumonia model infected with the colistin-susceptible strain. This lack of effectiveness with colistin monotherapy has been observed in other experimental models of infection with other pathogens, such as *Klebsiella pneumoniae* (Pachón-Ibáñez et al, 2018), and in a retrospective clinical study, in which suboptimal efficacy against Enterobacteriaceae infections was observed (de Oliveira et al, 2015). One explanation could be that the defined target $f\text{AUC}_{0-24}/\text{MIC}$ value used to predict colistin efficacy was based on achieving a 2 $\log_{10}$ decrease in bacterial concentrations in the lungs or thigh, but the mice survival rates were not evaluated (Cheah et al, 2015). In addition, the low efficacy in reducing bacterial concentration in the lungs and, therefore, in decreasing the mortality, could be due to the known poor colistin lung penetration (Hussain et al, 2020).

According to the in vitro studies, the colistin-susceptible Pa147 strain was more virulent than the colistin-resistant PaM1 strain, showing higher motility, better fitness and greater biofilm production. Moreover, the inoculum required to achieve 100% mortality and bacteraemia and similar bacterial concentrations in the lungs and blood was ~1 $\log_{10}$ higher with the colistin-resistant strain than the colistin-susceptible strain. These results are in accordance with other studies showing that the decrease in both the in vitro and in vivo fitness and virulence of *Acinetobacter baumannii* after acquisition of colistin resistance was due to a *pmrA* mutation (Lopez-Rojas et al, 2013). However, the results are comparable between the two strains in our study because we characterised the pneumonia model such that we could achieve the same mortality and bacterial concentrations in tissues with the colistin-susceptible and colistin-resistant strains.

The only clinical study evaluating the combination of antibiotics with a single dose of IgM-IG was carried out in patients with sepsis or septic shock by multidrug-resistant GNB infections, in which the group that received antibiotics in combination with IgM-IG had a reduction of the mortality rate compared with the group treated with antibiotics alone (Giamarellos-Bourboulis et al, 2016). The caveats of this clinical study are its retrospective design, with cases receiving IgM-IG at the discretion of attending physicians, and the data were collected from a database of intensive care unit-acquired sepsis from 63 study sites. Although 77% of patients had ventilator-associated pneumonia, only 21% were caused by *P. aeruginosa*. In this context, in the experimental *P. aeruginosa* pneumonia model, we found that the administration of IgM-IG

combined with colistin was useful in reducing the bacterial lung concentration, which points to the possible efficacy of antibiotics combined with IgM-IG to diminish the mortality of multidrug-resistant *P. aeruginosa* pneumonia in human beings, which may be as high as 33.3% in patients with ventilator-associated pneumonia treated with colistimethate sodium (Mogyoródi et al, 2022). To extrapolate these results to pneumonia by other multidrug-resistant GNB deserves further research with experimental pneumonia models by other pathogens.

Although our results suggest that the combined treatment of colistin to IgM-enriched immunoglobulin could improve lung infections by multidrug-resistant *P. aeruginosa*, we have to point out several limitations to the study. The 3R rules (Hubrecht & Carter, 2019) prevent us from increasing the number of strains in which to confirm the present results; however, to avoid strain-dependent results, we have used two different strains in terms of colistin susceptibility and virulence. Also, the 3R rules recommend to avoid other infection models to test the same hypothesis, as well as the increase of the sample size of the experimental groups, although it has not prevented to show better efficacy of the combination of IgM-IG with colistin, in the performed experiments. Finally, as in any animal model, a limitation is the general caution to translate the preclinical studies to the clinical setting, although the antibiotics dosages have been chosen according to the pharmacodynamics targets in human beings.

In summary, these data suggest that adjuvant treatment using iv IgM-enriched immunoglobulin together with colistin could improve outcomes in cases of treating multidrug-resistant *P. aeruginosa* infections. However, only controlled and randomized clinical studies can conclude whether the combination of IgM-IG and colistin reduces mortality in pneumonia caused by colistin-susceptible and multidrug-resistant *P. aeruginosa*.

# Materials and Methods

### Bacterial strains

Two clinical multidrug-resistant *P. aeruginosa* strains that cause BSI and have different colistin (COL) susceptibilities were used for inducing pneumonia in mice: Pa147 (MIC 0.5 mg/l) and PaM1 (MIC >128 mg/l). For biofilm assays, *A. baumannii* ATCC 19606 and *A. baumannii* CR17 (Lopez-Rojas et al, 2013) were used.

### *Animals*
Male immunocompetent C57BL/6J mice (20 g, 7–9 wk old) were used (Production and Experimentation Animal Centre, University of Seville). Animals had a sanitary status of murine pathogen free and were assessed for genetic authenticity. Mice were housed in an individually ventilated cage system under specific pathogen-free conditions, with ad libitum access to water and food. The study was carried out following the recommendations of the Guide for the Care and Use of Laboratory Animals [National Research Council, 2011] https://grants.nih.gov/grants/olaw/guide-for-the-care-and-use-of-laboratory-animals.pdf). In vivo experiments were approved by the Committee on the Ethics of Animal Experiments of the University Hospital Virgen del Rocío (0784-N-15) and the

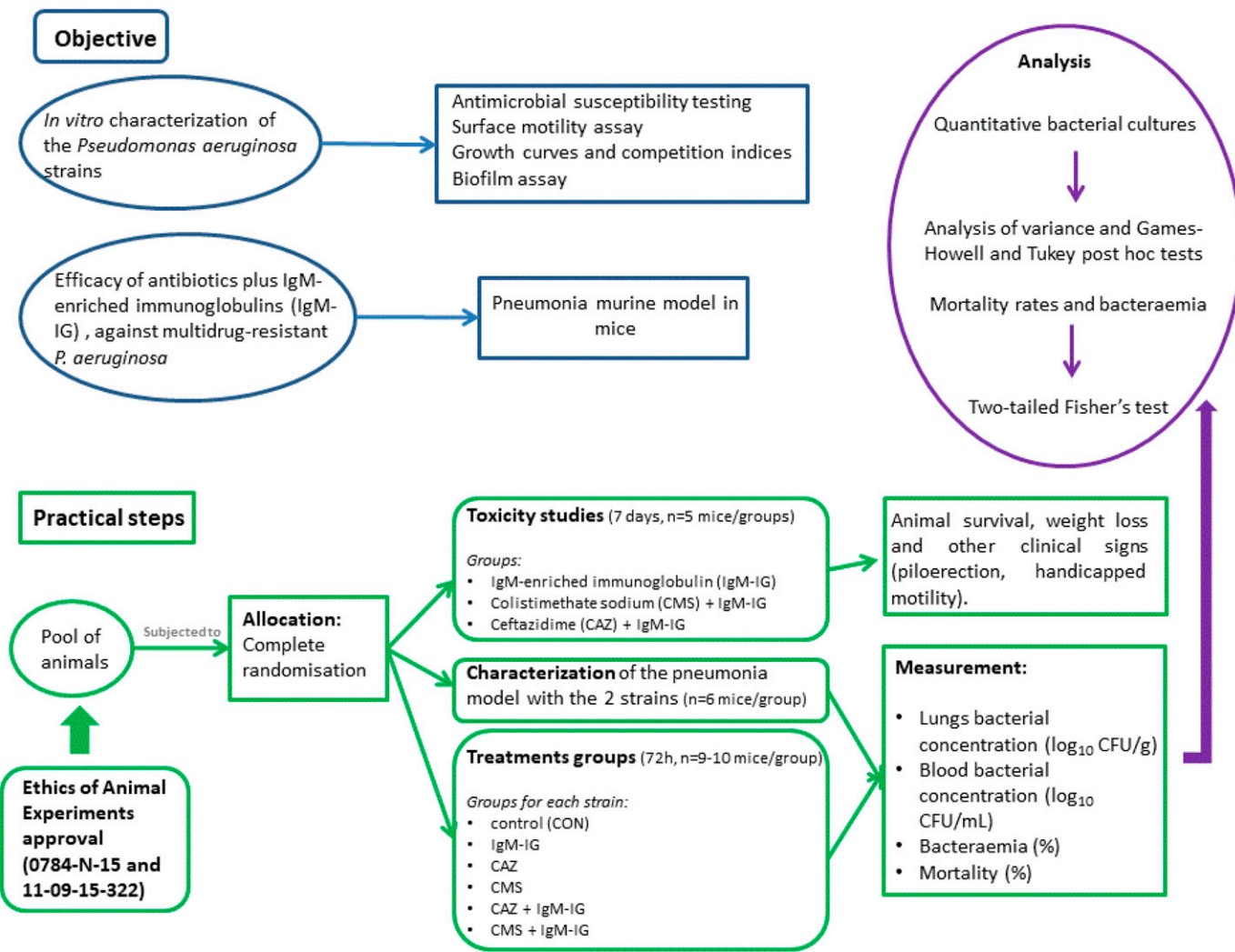

**Figure 6. Experimental study design.**

Ministry of Agricultura, Pesca y Desarrollo Rural (11-09-15-322). The animal model experimental study design is represented in Fig 6.

## Antibiotics and immunoglobulins

For the in vitro studies, antimicrobials were used as standard laboratory powders (Sigma-Aldrich). For the efficacy studies, clinical formulations were used: colistimethate sodium (CMS, Promixin, 80 mg), ceftazidime (CAZ, Normon S.A, 500 mg), and IgM-IG (Pentaglobin, 50 mg/ml).

## Antimicrobial susceptibility testing

MICs were determined using the broth microdilution method following the EUCAST recommendations and interpreted according to EUCAST breakpoints (The European Committee on Antimicrobial Susceptibility Testing, 2022) (https://www.eucast.org/clinical_breakpoints/). MBCs were determined by subculturing onto 100 µl of antimicrobial-free Mueller–Hinton agar in wells containing antimicrobial concentrations greater than or equal to the MIC of the corresponding agent. Studies were performed in triplicate.

## Surface motility assay

Surface motility was measured as previously described (LewisOscar et al, 2018). Briefly, overnight cultures of each strain were adjusted in PBS to an OD of 0.6 (600 nm) (Lonza). Bacterial suspension (3 µl) was plated onto LB containing 0.3% agarose. Plates were incubated for 24 h at 37°C with 80% humidity, and the radios of surface extensions were measured. Assays were performed in triplicate.

## In vitro growth curves and competition indices

Each strain ($5 \times 10^5$ CFU/ml) was grown in 10 ml of MHBII (Sigma-Aldrich) at 37°C. Then, at different time points (2, 4, 8, and 24 h), 100 µl aliquots were taken, and twofold dilutions were plated on blood agar plates. Competitive growth between the two strains was assessed in MHB by mixing $5 \times 10^5$ CFU/ml of each strain in the same

culture. After 24 h, aliquots from the cultures were seeded on both blood agar and Mueller–Hinton plates with 2 mg/l colistin to differentiate the colonies (Lopez-Rojas et al, 2013), as whereas both strains grow on blood agar plates, only colistin-resistant PaM1 grow on plates containing colistin. Competition indices were defined as the number of recovered CFUs of PaM1/the number of recovered CFUs of the Pa147 strain, divided by the number of CFUs in the PaM1 inoculum/the number of CFUs in the Pa147 inoculum. If no colonies were recovered from cultures, the limit of detection of the assay (1 CFU) was used to calculate competition indices. Experiments were performed in triplicate.

### Biofilm assay

Biofilm production was measured as previously described (O'Toole et al, 1999). *A. baumannii* ATCC 19606 and AbCR17 were used as positive and negative controls, respectively. Strains were cultured in 20 ml of MHB overnight at 37°C and subsequently diluted to $10^6$ CFU/ml in MHB. A total of 200 $\mu$l of the cell suspension was added to each well of a round-bottom 96-well plate and grown overnight at 37°C. Each well was washed twice to remove nonadherent bacteria, and 200 $\mu$l of 0.4% crystal violet dye (Sigma-Aldrich) was added to each well. After 10-min incubation at room temperature, the wells we washed twice and 200 $\mu$l of 96% ethanol added to each. After 15 min of incubation at room temperature, biofilm formation was quantified by measuring the OD at 580 nm (Asys UVM 340 Microplate Reader, EE.UU.). Assays were performed in triplicate. The results are represented with normalisation to the positive control strain, which was taken as 100%.

### Toxicity studies

Before performing the efficacy studies, all therapies were evaluated in five healthy male immunocompetent C57BL/6J mice (20 g), which were treated with the dosage and regimen to be used. Mice were evaluated for 7 d, and no weight loss or changes in motility and/or mortality were observed.

### Efficacy studies

A previously characterised murine pneumonia model was used (Gil-Marques et al, 2018). Firstly, minimal lethal dose (MLD, concentration of inoculum killing 100% of the animals) was determined using the Reed and Munch method (O'Reilly & Squires, 1996). Briefly, groups of six mice were inoculated intratracheally with 50 $\mu$l of decreasing concentrations of inoculum of each strain, beginning with 10.5 $\log_{10}$ CFU/ml, and animals were observed for 7 d. Then, groups of mice were ip anesthetised (ketamine/diazepam) and intratracheally infected with 50 $\mu$l of the MLD: 8.59 and 9.66 $\log_{10}$ CFU/ml for Pa147 and PaM1 strains, respectively. Then, mice were randomly assigned to one of the following groups: (i) control, infected and untreated; (ii) CMS (20 mg/kg/q8h/ip); (iii) CAZ (100 mg/kg/q12h/ip); (iv) IgM-IG (430 mg/kg/iv), single dose administered 30 min post-infection; (v) IgM-IG plus CMS; and (vi) IgM-IG plus CAZ. Antibiotic treatments were initiated 4 h post-infection and lasted 72 h. Antibiotics dosages were based on previous PK/PD data and their proven efficacy in experimental models of infection (Chen et al, 2018; Pachón-Ibáñez et al, 2018). IgM-IG dosage was based on of

Giamarellos-Bourboulis et al (2016) study on patients with sepsis and septic shock (Giamarellos-Bourboulis et al, 2016).

Mortality was recorded for 72 h after beginning the treatments. Immediately after animal death, or on euthanasia at the end of the experiment (sodium thiopental, ip), lungs and blood samples were aseptically obtained and processed for quantitative cultures ($\log_{10}$ CFU/g and $\log_{10}$ CFU/ml). Blood samples were also studied for qualitative cultures.

### Statistical analysis

Mortality rates and positive qualitative blood cultures are expressed as percentages, and differences between groups were compared using the two-tailed Fisher's test. Quantitative bacterial cultures in the lungs ($\log_{10}$ CFU/g), expressed as means ± SEM, were compared by Tukey's post hoc tests, after confirming the homogeneity of variance between groups using Levene's test ($P$ = 0.219 and 0.091, for Pa147 and PaM1, respectively). For the quantitative blood cultures ($\log_{10}$ CFU/ml), the Games–Howell test post hoc tests was used (Levene's test, $P$ = 0.000 and 0.000, for Pa147 and PaM1, respectively). As the hypothesis of the study was that combining IgM-IG with antibiotics improves their therapeutic efficacy, in the primary analysis we compared each combination with the corresponding monotherapies and controls. Moreover, as a secondary analysis, we made the comparison between the CMS and CAZ treatment groups. Finally, the eta-squared for treatment comparisons was calculated. $P$ < 0.05 was considered to indicate significance. SPSS v24.0 software was used (SPSS Inc.).

# Data Availability

All the study data is available.

### Animal experiments approval statement

Ethical approval for this study was obtained from the Committee on the Ethics of Animal Experiments of the University Hospital Virgen del Rocío (0784-N-15) and the Ministry of Agricultura, Pesca y Desarrollo Rural (11-09-15-322).

The present study followed the recommendations of the Guide for the Care and Use of Laboratory Animals (https://grants.nih.gov/grants/olaw/guide-for-the-care-and-use-of-laboratory-animals.pdf).

# Supplementary Information

# Acknowledgements

This study was supported by the Consejería de Salud of the Junta de Andalucía (PI-0622-2012) and by Plan Nacional de I+D+i 2013-2016 and Instituto de Salud Carlos III, Subdirección General de Redes y Centros de

Investigación Cooperativa, Ministerio de Economía, Industria y Competitividad, Spanish Network for Research in Infectious Diseases (REIPI RD16/0016/0009)—co-financed by European Development Regional Fund A way to achieve Europe, Operative program Intelligent Growth 2014-2020. T Cebrero-Cangueiro is supported by the V Plan Propio of the University of Seville with a postdoctoral contract as research personnel in training. G Labrador-Herrera has a grant from the Ministerio de Ciencia, Innovación y Universidades, Instituto de Salud Carlos III, cofinanced by the European Development Regional Fund (A Way to Achieve Europe). ME Pachón-Ibáñez is supported by program "Nicolás Monardes" (C1-0038-2019), Servicio Andaluz de Salud, Junta de Andalucía, Spain.

## Author Contributions

T Cebrero-Cangueiro: formal analysis, investigation, and writing—original draft.

G Labrador-Herrera: formal analysis and investigation.

M Carretero-Ledesma: investigation.

S Herrera-Espejo: investigation.

R Álvarez-Marín: formal analysis, supervision, and writing—review and editing.

J Pachón: conceptualization, formal analysis, supervision, validation, visualization, and writing—review and editing.

JM Cisneros: formal analysis, supervision, and writing—review and editing.

ME Pachón-Ibáñez: conceptualization, formal analysis, supervision, funding acquisition, validation, visualization, methodology, project administration, and writing—review and editing.

## Conflict of Interest Statement

The authors declare that they have no conflict of interest.

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
