## [Reviewer comments · Life Science Alliance]

Life Science Alliance

IgM-enriched immunoglobulin improves colistin efficacy in pneumonia model by *Pseudomonas aeruginosa*

Tania Cebrero-cangueiro, Gema Labrador-Herrera, Marta Carretero-Ledesma, Soraya Herrera-Espejo, Rocío Álvarez-Marín, Jerónimo Pachón, Jose Miguel Cisneros, and María Eugenia Pachon Ibañez

DOI: <https://doi.org/10.26508/lsa.202101349>

Corresponding author(s): *María Eugenia Pachon Ibañez, Institute of Biomedicine of Seville*

Review Timeline:

Submission Date:	2021-12-23
Editorial Decision:	2022-01-31
Revision Received:	2022-04-11
Editorial Decision:	2022-05-02
Revision Received:	2022-05-30
Editorial Decision:	2022-06-01
Revision Received:	2022-06-07
Accepted:	2022-06-07

Scientific Editor: Novella Guidi

Transaction Report:

January 31, 2022

Re: Life Science Alliance manuscript #LSA-2021-01349-T

Dr. María Eugenia Pachon Ibañez
Institute of Biomedicine of Seville
Infectious Diseases
Avda Manuel Siurot s/n
Seville, Seville 41013
Spain

Dear Dr. Pachon Ibañez,

Thank you for submitting your manuscript entitled "IgM-enriched immunoglobulin improves colistin efficacy in a pneumonia model by *Pseudomonas aeruginosa*" to Life Science Alliance. The manuscript was assessed by expert reviewers, whose comments are appended to this letter. We, thus, encourage you to submit a revised version of the manuscript back to LSA that responds to all of the reviewers' points.

Thank you for this interesting contribution to Life Science Alliance. We are looking forward to receiving your revised manuscript.

Sincerely,

B. MANUSCRIPT ORGANIZATION AND FORMATTING:

Reviewer #1 (Comments to the Authors (Required)):

In this study, the authors sought to evaluate the efficacy of combining IgM-enriched immunoglobulin with two antibiotics (ceftazidime and colistin) in mice that were intratracheally inoculated with multidrug-resistant *Pseudomonas aeruginosa* strains. The study objective is very interesting, although there are several shortcomings of the descriptions in the paper and analyses that may not allow the conclusions to be fully supported. These need to be clarified and the authors need to write their paper according to ARRIVE 2.0 guidelines to guarantee that the study can be fully reproducible and able to be adequately interpreted. Also, this study does not provide a good overview of the research subject and does not adequately discuss future directions for research according to the existing literature. These are my main comments:

Major comments:

1. The introduction failed to justify why IgM-Ig could be useful combined with antibiotics could be useful to treat MDR bacterial infections. Even when the authors mentioned some studies testing this combination, they did not cover sufficiently the potential mechanisms why they might work. This is a key aspect to reflect the translational viewpoint of this study that the authors were seemingly trying to suggest throughout the manuscript.
2. This study involved a complex animal model experimental study design due to the multiple treatment groups and bacterial strains involved. Please add a visual representation of the flow of the study which would be your Figure 1, since this is what is recommended by the ARRIVE 2.0 guidelines (<https://dx.plos.org/10.1371/journal.pbio.3000411>). You may use the Experimental Design Assistant (<https://doi.org/10.1371/journal.pbio.2003779>) to build this figure.
3. The authors are trying to test if adding IgM-Ig to antibiotic treatment is efficacious in an animal model. Antibiotics are known to be efficacious, whereas IgM-Ig efficacy is unknown. Nonetheless, the authors present their analyses and results as if IgM-Ig efficacy had already been proven, and as if they wanted to test if antibiotics can be efficacious, too. According to the study objective, the authors should have included comparisons (with their corresponding statistical test) of:
 - a. CMS+IgM vs CON, CMS+IgM vs IgM, CMS+IgM vs CMS, and CMS+IgM vs CAZ
 - b. CAZ+IgM vs CON, CAZ+IgM vs IgM, CAZ+IgM vs CMS, and CAZ+IgM vs CAZ
 - c. The comparison of CMS+IgM vs CAZ+IgM would be in a secondary order of importance after efficacy in the prior analyses has been shown.
4. Regarding comment number 3, the adequate statistical test would be Tukey test. However, homoscedasticity would need to be verified. If it is not present, Welch's test should be used instead of ANOVA and the Games-Howel test should be used instead of Tukey test. Please show the results for Leven test in the manuscript to justify that comparisons are correct due to the small number in each group.
5. The authors mention having compared bacteremia and mortality by Fisher's test but mention that there are differences with respect to baseline. This statistical test cannot be used to show differences with respect to X group. Please clarify how you reached this conclusion.
6. The correct comparisons would have been to compare the main intervention group CAZ+IgM against all other groups, as well as CMS+IgM against all other groups.
7. Please calculate eta-squared for treatment comparisons to support your conclusions.
8. Please review the full ARRIVE 2.0 Explanation and Elaboration document (<https://dx.plos.org/10.1371/journal.pbio.3000411>) and make sure that you have adequately reported all elements. Any elements missing need to be added or further explained within the manuscript. When you have finished adding all elements, please go to <https://www.goodreports.org/reporting-checklists/arrive2/> to fill in the checklist with the pages of the manuscript where all elements can be found and upload this checklist as a PDF for peer-review-only as supplementary material alongside your revised manuscript.
9. The discussion section of the paper is also lacking the translational viewpoint. Please mention what potential mechanisms may underly your findings since this discussion could be very important to shape the direction of future studies.
10. Limitations of the study need to be added in the discussion in a new paragraph.

Minor comments:

1. Lines 39-40: This statement is misleading. Even when *pseudomonas aeruginosa* may cause community-acquired infections in some patients, infections in ICU patients are more frequently hospital-acquired infections. Please correct.
2. Lines 42-43: It is not clear what you mean with this phrase, please rephrase it: "This microorganism is a paradigm of

antimicrobial resistance, which continues to increase globally (3)."

3. Lines 47-49: This also needs to be rewritten to improve clarity of what you mean: "Although several antibiotics have recently been approved for human use (6), they cannot be produced at a rate equivalent to that in which antibiotic resistance is acquired (7)."

4. Line 53: Reference no. 9 does not support this statement since the study you are citing did not evaluate the prognosis of human infections.

5. Line 58: Please be more specific on what you mean by "antibiotic pressure".

6. Lines 58-59: Reference no. 11 does not support this statement since this is a pharmacoepidemiological study not evaluating antibiotic pressure nor modulation of the immune system.

7. Lines 58-59: Reference no. 12 does not support this statement since the authors of the study you are citing performed experiments in mice, not in patients as you are suggesting.

8. Lines 58-59: Reference no. 13 does not support this statement since the study you are citing did not evaluate protection against reinfection.

9. Lines 59-61: Reference no. 9 does not support this statement since the study you are citing did not evaluate efficacy nor effectiveness of IVIg in combination with antibiotics on patients with severe infections of different etiologies.

10. Lines 63-67: Reference no. 15 was an observational study. Observational studies cannot be used to conclude that "IgM-Ig with antibiotics resulted in increased survival of patients...". Please rewrite to avoid suggesting a cause-effect conclusion not supported by the study you are citing.

11. Lines 63-67: Please also eliminate the following: "However, despite these results, this treatment has not been introduced to clinical practice."

12. Lines 68-71: This paragraph should say "antibiotics plus IgM-enriched immunoglobulins" instead of "IgM-enriched immunoglobulins in combination with antibiotics" since the efficacy of antibiotics is already known and you are trying to show that adding IgM-Ig may be efficacious, not the other way around. Please also change all other descriptions throughout the entire manuscript to comply with this order of ideas, including figures and tables.

13. Lines 68-71: Please mention the specific antibiotics you tested.

14. Please include standard error instead of standard deviation since it is more useful to interpret your analyses.

15. Lines 122-124: You mention "several studies" but only cite 1 study. Please correct to mention that this has only been shown in that specific study.

16. Paragraph starting in line 147: Reference number 15 included patients and was observational, not experimental. Your study was performed in animal models. Please contextualize these important differences between the studies.

Reviewer #3 (Comments to the Authors (Required)):

In this paper by Cebrero-Cangueiro et. al. they describe the use of IgM enriched immunoglobulin alone and in concert with two antibiotics as a treatment in a *P. aeruginosa* pneumonia model. They find that IgM enriched immunoglobulin used in concert with colistin leads to reduced bacterial burden and less mortality than colistin alone. The experiments are well controlled and the research findings are important and interesting. Unfortunately the study is a little underpowered, so despite showing large decreases in bacterial burden in the blood and mortality, only the lung results (Pa147) and mortality (PaM1) are significant. Major comments:

-As I mentioned above a lot of the data shows a drop in bacterial numbers/ mortality that is not significant. Increasing the numbers of mice may lead to significant results, however the authors generally do well to not over-analyse the results as they are. However the abstract is misleading to state 'IgM-Ig plus colistin reduced the bacterial concentration of the colistin-susceptible strain in the lungs and blood' when the blood reduction was not found to be significant.

-Discussion of mechanism of IgM-enriched immunoglobulin is lacking and needed. Does any literature shed light on why it may have a significant effect in the lungs but not blood? Is the lack of serum/ complement killing in C57BL/6 mice a reason for the differences between animal studies and human trials? What is the mechanism for IgM-Ig improving antibiotic susceptibility in general?

-Would results in table 2 (the lung/ blood part) be easier to digest as a graph (or two)? It makes finding the significant results a lot easier.

Minor comments:

Line 39: *Pseudomonas* is also a major cause of hospital-acquired infections

Results: The purpose of the motility, biofilm and competitive experiments is unclear. Is this just to look at virulence traits? Do these effect the interpretation of the IgM-Ig results? Some clarification of why these are being done in the manuscript would be useful.

Table 2- Needs to label Lung and blood as 'Log10 CFU/g' not just 'CFU/g'

Reviewer #1 (Comments to the Authors (Required)):

In this study, the authors sought to evaluate the efficacy of combining IgM-enriched immunoglobulin with two antibiotics (ceftazidime and colistin) in mice that were intratracheally inoculated with multigrug-resistant Pseudomonas aeruginosa strains. The study objective is very interesting, although there are several shortcomings of the descriptions in the paper and analyses that may not allow the conclusions to be fully supported. These need to be clarified and the authors need to write their paper according to ARRIVE 2.0 guidelines to guarantee that the study can be fully reproducible and able to be adequately interpreted. Also, this study does not provide a good overview of the research subject and does not adequately discuss future directions for research according to the existing literature. These are my main comments:

Regarding to the ARRIVE guidelines 2.0: author checklist, we do have it but did not send it, as it was not required when we submitted the manuscript. As requested we are sending it now.

Major comments:

1. The introduction failed to justify why IgM-IG could be useful combined with antibiotics could be useful to treat MDR bacterial infections. Even when the authors mentioned some studies testing this combination, they did not cover sufficiently the potential mechanisms why they might work. This is a key aspect to reflect the translational viewpoint of this study that the authors were seemingly trying to suggest throughout the manuscript.

The reviewer is right in his/her statement that “we did not cover sufficiently the potential mechanisms why they might work”. To address this issue we have added in the Introduction the following paragraph: “IgG is the most abundant antibody and provides protection against infections, bacterial and/or viral. Among its most important functions are to neutralize and eliminate the pathogen, in this case *P. aeruginosa*, toxins

generated by the bacteria, and/or other substances produced in inflammatory processes or cell destruction (Birdsall and Casadevall. 2020). Nevertheless, IgG can take some time to form after infection, so combining it with IgM, first line of defense against infection by inhibiting bacterial adhesion to epithelial cells and neutralizing toxins, could increase the immune response and be an adjuvant to antibiotics (Birdsall and Casadevall. 2020). These reasons suggest that using IgM-IG could modulate the immune system of the mice increasing the efficacy of the tested antibiotics.”

In addition, in the previous manuscript, we referred the following data from Rossmann *et al.* (reference 14 of the previous manuscript) “IgM combined with IgG and IgA have showed opsonophagocytic activity against carbapenem-resistant *P. aeruginosa* (Rossmann *et al.* 2015)”, as potential mechanisms of why IgM-IG might work.

These data have been included in the same paragraph of the revised manuscript. (lines 57-69)

2. This study involved a complex animal model experimental study design due to the multiple treatment groups and bacterial strains involved. Please add a visual representation of the flow of the study which would be your Figure 1, since this is what is recommended by the ARRIVE 2.0 guidelines (<https://dx.plos.org/10.1371/journal.pbio.3000411>). You may use the Experimental Design Assistant (<https://doi.org/10.1371/journal.pbio.2003779>) to build this figure.

We have added it as requested by the Reviewer.

3. The authors are trying to test if adding IgM-IG to antibiotic treatment is efficacious in an animal model. Antibiotics are known to be efficacious, whereas IgM-IG efficacy is unknown. Nonetheless, the authors present their analyses and results as if IgM-IG efficacy had already been proven, and as if they wanted to test if antibiotics can be efficacious, too. According to the study objective, the authors should have included comparisons (with their corresponding statistical test) of:

- a. CMS+IgM vs CON, CMS+IgM vs IgM, CMS+IgM vs CMS, and CMS+IgM vs CAZ**
- b. CAZ+IgM vs CON, CAZ+IgM vs IgM, CAZ+IgM vs CMS, and CAZ+IgM vs CAZ**
- c. The comparison of CMS+IgM vs CAZ+IgM would be in a secondary order of importance after efficacy in the prior analyses has been shown.**

As specified in the last paragraph of the introduction, the aim of this study was to evaluate the efficacy of IgM-enriched immunoglobulins (IgM-IG) as an adjuvant with antibiotics regularly used in treating pneumonia by MDR *P. aeruginosa*, not the IgM-IG alone. This is the reason why we only compared the groups with IgM-IG alone or plus antibiotics with their respective monotherapies, as the primary analysis (line 307).

However, as requested by the Reviewer, as a secondary analysis, we have compared CMS + IgM-IG vs. CAZ + IgM-IG, CMS + IgM-IG vs. CAZ, and CAZ + IgM-IG vs. CMS, in the pneumonia by the two strains (added in Table 2) and (lines 309).

4. Regarding comment number 3, the adequate statistical test would be Tukey test. However, homoscedasticity would need to be verified. If it is not present, Welch's test should be used instead of ANOVA and the Games-Howel test should be used instead of Tukey test. Please show the results for Leven test in the manuscript to justify that comparisons are correct due to the small number in each group.

The Reviewer is right. Thus, in the previous manuscript, we performed the post hoc Tukey test only after checking the homogeneity of variance between groups with the Levene test in the quantitative analysis in lungs. Nevertheless, when there was no homogeneity of variance the Dunnett's T3 was used. As the Reviewer states, Games-Howel test is one of the multiple comparisons tests that do not assume equal variances, among others such as Dunnett's T3, Tamhane's T2, and Dunnett's C.

However, regarding this comment of the Reviewer, we have modified the Statistical Analysis section of the manuscript to make it clearer, and have added the Levene results as requested.

5. The authors mention having compared bacteremia and mortality by Fisher's test but mention that there are differences with respect to baseline. This statistical test cannot be used to show differences with respect to X group. Please clarify how you reached this conclusion.

We interpret that with the sentence “to show differences with respect to X group” the Reviewer means that the bacteremia and mortality analysis were performed with multiple groups comparisons. This is not the case; we compare these data with each treatment group vs. the controls or the other groups, separately, in 2x2 tables.

6. The correct comparisons would have been to compare the main intervention group CAZ+IgM against all other groups, as well as CMS+IgM against all other groups.

We understand that this comment is the same detailed in the # 3. As requested by the Reviewer, as a secondary analysis, we have compared the CMS + IgM-IG vs. the CAZ + IgM-IG groups, the CMS + IgM-IG vs. CAZ, and CAZ + IgM-IG vs. CMS in the pneumonia by the two strains (added in Table 2) and (lines 309).

7. Please calculate eta-squared for treatment comparisons to support your conclusions.

Following the Reviewer request we have calculated eta-squared for treatment comparisons. We have added this information in the Statistical Analysis section (line 310) and in the Results section (lines 136-138).

8. Please review the full ARRIVE 2.0 Explanation and Elaboration document (<https://dx.plos.org/10.1371/journal.pbio.3000411>) and make sure that you have adequately reported all elements. Any elements missing need to be added or further explained within the manuscript. When you have finished adding all elements, please

go to <https://www.goodreports.org/reporting-checklists/arrive2/> to fill in the checklist with the pages of the manuscript where all elements can be found and upload this checklist as a PDF for peer-review-only as supplementary material alongside your revised manuscript.

As previously commented, regarding to the ARRIVE guidelines 2.0: author checklist, we do have it but did not send it, as it was not required when we submitted the manuscript. As requested we are sending it now.

9. The discussion section of the paper is also lacking the translational viewpoint. Please mention what potential mechanisms may underly your findings since this discussion could be very important to shape the direction of future studies.

Following the request of the Reviewer, we have added in the Discussion Section the sentence “, which support the combination of IgM-IG when colistin is the last resort in the treatment of pneumonia by multidrug-resistant *P. aeruginosa*.”

10. Limitations of the study need to be added in the discussion in a new paragraph.

As requested by the Reviewer, we have added a new paragraph in the Discussion Section with the limitations of the study: “The limitations of the study are the general caution to translate the preclinical studies to the clinical setting, although the antibiotics dosages have been chosen according to the pharmacodynamics targets in human beings. Moreover, the 3R rules (<https://caat.jhsph.edu/principles/chap4d>) preclude the increase of the sample size of the experimental mice groups, although it has not prevented to show the best efficacy of the combination of IgM-IG with colistin.”

Minor comments:

1. Lines 39-40: This statement is misleading. Even when pseudomonas aeruginosa may cause community-acquired infections in some patients, infections in ICU patients are more frequently hospital-acquired infections. Please correct.

To clarify this sentence, attending the suggestion of the Reviewer, we have modified it to: “community- and hospital-acquired infections, especially in intensive care units (ICUs)”. (lines 39-40).

2. Lines 42-43: It is not clear what you mean with this phrase, please rephrase it: "This microorganism is a paradigm of antimicrobial resistance, which continues to increase globally (3)."

Following the request of the Reviewer, we have rephrased this sentence to: “This microorganism is one of the paradigms of the global threat of antimicrobial resistance.” (line 43).

3. Lines 47-49: This also needs to be rewritten to improve clarity of what you mean: "Although several antibiotics have recently been approved for human use (6), they cannot be produced at a rate equivalent to that in which antibiotic resistance is acquired (7)."

Following the suggestion of the Reviewer, we have modified the sentence to make it clearer "Although in the last years several antibiotics have recently been approved for human use (Lupia et al. 2020), new antimicrobials cannot be produced by industry and approved for their clinical use at a rate equivalent to that in which antibiotic resistance is acquired (Pachori et al. 2019)." (lines 48-52).

4. Line 53: Reference no. 9 does not support this statement since the study you are citing did not evaluate the prognosis of human infections.

The Reviewer is right; we have modified it as requested, adding the following references: Schedel et al. 1991; Werdan et al. 1996; Laupland et al. 2007; Kreymann et al. 2007.

5. Line 58: Please be more specific on what you mean by "antibiotic pressure".

We mean the exposure of bacteria to antibiotics, which in human's beings, as well as in animals and in the environment is one of the main factors responsible for antibiotic resistance development. In any case, we have removed this sentence in the revised manuscript, following the Comment # 1 of the Reviewer.

6. Lines 58-59: Reference no. 11 does not support this statement since this is a pharmacoepidemiological study not evaluating antibiotic pressure nor modulation of the immune system.

The Reviewer is right; as we have detail in the previous minor Comment, we have removed this reference, as well the all sentence.

7. Lines 58-59: Reference no. 12 does not support this statement since the authors of the study you are citing performed experiments in mice, not in patients as you are suggesting.

The Reviewer is right; we have corrected the sentence as requested.

8. Lines 58-59: Reference no. 13 does not support this statement since the study you are citing did not evaluate protection against reinfection.

The Reviewer is Right; we have corrected the mistake as requested.

9. Lines 59-61: Reference no. 9 does not support this statement since the study you

are citing did not evaluate efficacy nor effectiveness of IVIg in combination with antibiotics on patients with severe infections of different etiologies.

The Reviewer is right; we have corrected the reference as requested and modified the sentence to make it clearer “A prospective, randomized clinical trial showed that antibiotics in combination with intravenous polyclonal immunoglobulin (Ig) preparation containing IgG, IgM, and IgA as an adjunctive therapy were associated with significantly improved survival in patients with septic shock (Schedel et al. 1991)”

10. Lines 63-67: Reference no.15 was an observational study. Observational studies cannot be used to conclude that "IgM-IG with antibiotics resulted in increased survival of patients...". Please rewrite to avoid suggesting a cause-effect conclusion not supported by the study you are citing.

The study of the reference no. 15 had a case-control design.

Following the Reviewer suggestion, we have modified the sentence as follows:

“Giamarellos-Bourboulis *et al.* (15), using a case-control design, have reported that a combination of antibiotics plus polyclonal IgM-enriched immunoglobulins (IgM-IG) increased survival of patients with sepsis or septic shock caused by multidrug-resistant GNBs, including *P. aeruginosa*.”

11. Lines 63-67: Please also eliminate the following: "However, despite these results, this treatment has not been introduced to clinical practice."

Following the Reviewer suggestion, we have eliminated this sentence.

12. Lines 68-71: This paragraph should say "antibiotics plus IgM-enriched immunoglobulins" instead of "IgM-enriched immunoglobulins in combination with antibiotics" since the efficacy of antibiotics is already known and you are trying to show that adding IgM-IG may be efficacious, not the other way around. Please also change all other descriptions throughout the entire manuscript to comply with this order of ideas, including figures and tables.

Following the Reviewer request, we have changed "IgM-enriched immunoglobulins in combination with antibiotics" to "antibiotics plus IgM-enriched immunoglobulins", through the entire manuscript.

13. Lines 68-71: Please mention the specific antibiotics you tested.

Following the request, we have added this information. (Line 86).

14. Please include standard error instead of standard deviation since it is more useful to interpret your analyses.

We have modified it, as requested (line 301) and Table 2 and Figure 4.

15. Lines 122-124: You mention "several studies" but only cite 1 study. Please correct to mention that this has only been shown in that specific study.

The Reviewer is right. We have added other references, as requested (Pachón-Ibanez et al. 2010; Parra-Millan et al. 2016; Docobo-Perez et al. 2012; Cai et al. 2018).

16. Paragraph starting in line 147: Reference number 15 included patients and was observational, not experimental. Your study was performed in animal models. Please contextualize these important differences between the studies.

Following the request of the Reviewer, to contextualize this comparison we have added, at the beginning of this paragraph, the following: "Our preclinical experimental results may be compared only with an observational case-control study in humans (Giamarellos-Bourboulis et al. 2016), because of the lack of data from other *in vivo* experimental or clinical studies. With the cautions of matching them, our...." (Lines 177-179).

Reviewer #3 (Comments to the Authors (Required)):

In this paper by Cebrero-Cangueiro et. al. they describe the use of IgM enriched immunoglobulin alone and in concert with two antibiotics as a treatment in a P. aeruginosa pneumonia model. They find that IgM enriched immunoglobulin used in concert with colistin leads to reduced bacterial burden and less mortality than colistin alone. The experiments are well controlled and the research findings are important and interesting. Unfortunately the study is a little underpowered, so despite showing large decreases in bacterial burden in the blood and mortality, only the lung results (Pa147) and mortality (PaM1) are significant.

Major comments:

-As I mentioned above a lot of the data shows a drop in bacterial numbers/ mortality that is not significant. Increasing the numbers of mice may lead to significant results, however the authors generally do well to not over-analyse the results as they are. However the abstract is misleading to state 'IgM-IG plus colistin reduced the bacterial concentration of the colistin-susceptible strain in the lungs and blood' when the blood reduction was not found to be significant.

As the Reviewer states, although the combination of IgM enriched immunoglobulin with antibiotics improves the clearance of the lung bacterial concentration significantly, compared with untreated mice and colistin monotherapy, the results are variable depending on the strain. Yet, the use of the combination statistically reduces the lung bacterial concentration for both isolates, independently in their colistin susceptibility. Also, as noted by the Reviewer, the combination is able to reduce the blood bacterial concentration compared with the untreated controls in 2.52 and 3.01 log₁₀ CFU/mL for the Pa147 (colistin susceptible) and the PaM1 (colistin resistant), respectively. Our experience in animal models of infection, make us think that increasing the mice to 15 per group, would have made this results significant, as this sample size is sufficient to detect differences of 1.5 log₁₀ between groups, with an error $\alpha = 0.05$ and an error $\beta = 0.20$. Notwithstanding, we believe that the reduction in blood is enough to observe the decrease in blood bacterial concentration without sacrificing a larger number of animals.

Nevertheless, to avoid misleading as noted by the Reviewer, we have modified this sentence, as follows: “Colistin plus IgM-IG reduced the bacterial lung concentrations of both colistin-susceptible and resistant strains (-2.91 and -1.73 log₁₀ CFU/g, respectively) and the bacteremia rate of the colistin-resistant strain (-44%).” (Lines 32-35).

Discussion of mechanism of IgM-enriched immunoglobulin is lacking and needed. Does any literature shed light on why it may have a significant effect in the lungs but not blood? Is the lack of serum/ complement killing in C57BL/6 mice a reason for the differences between animal studies and human trials? What is the mechanism for IgM-IG improving antibiotic susceptibility in general?

Regarding the mechanism of IgM-enriched immunoglobulin therapeutic efficacy, in the Introduction of the manuscript we referred the following data from Rossmann *et al.* (reference 14 of the previous manuscript): “IgM combined with IgG and IgA have showed opsonophagocytic activity against carbapenem-resistant *P. aeruginosa*”. Also, to respond to the Comment 1 of the Reviewer 1, we added in the revised manuscript: “IgG is the most abundant antibody and provides protection against infections, bacterial and/or viral. Among its most important functions are to neutralize and eliminate the pathogen, in this case *P. aeruginosa*, toxins generated by the bacteria, and/or other substances produced in inflammatory processes or cell destruction (Birdsall and Casadevall. 2020). Nevertheless, IgG can take some time to form after infection, so combining it with IgM, first line of defense against infection by inhibiting bacterial adhesion to epithelial cells and neutralizing toxins, could increase the immune response and be an adjuvant to antibiotics (Birdsall and Casadevall. 2020). These reasons suggest that using IgM-IG could modulate the immune system of the mice increasing the efficacy of the tested antibiotics. Other approaches to immunomodulation, different to immunoglobulins, have also been reported.” These data have been included in the same paragraph of the revised manuscript. (lines 57-67).

We do not know data from the literature about different effects of IgM-enriched immunoglobulin in the lungs *vs.* blood. Because of the primary infection of our experimental model is the lungs, with bacterial burden higher than in blood, we hypothesize that the therapeutic effect of immunoglobulins is primarily directed to the infection source.

As if the lack of serum/complement killing in C57BL/6 mice is a reason for the differences between animal studies and human trials, our results show that colistin plus IgM-IG improves the results of colistin alone against *P. aeruginosa*. These results are not contradictory with those from the case-control study in human beings of Giamarellos-Bourboulis *et al.* (reference 15 of the previous manuscript).

Finally, we do not believe that IgM-IG improves the antibiotic susceptibility in general. We think that its effect is as a therapeutic adjuvant to antibiotics.

Would results in table 2 (the lung/ blood part) be easier to digest as a graph (or two)? It makes finding the significant results a lot easier.

As suggested by the Reviewer, we have included the lungs and blood bacterial concentrations results in Figures 4 and 5.

Minor comments:

Line 39: Pseudomonas is also a major cause of hospital-acquired infections

As suggested by the Reviewer, it has been modified in the revised manuscript (Line 40).

Results: The purpose of the motility, biofilm and competitive experiments is unclear. Is this just to look at virulence traits? Do these effect the interpretation of the IgM-IG results? Some clarification of why these are being done in the manuscript would be useful.

The purpose of the motility, fitness and biofilm production analysis were effectively done to know if the two strains had the same virulence traits and, consequently, to use specific inoculum size with each strain to achieve 100% mortality and bacteraemia, as well as similar bacterial concentrations in lungs and blood in the control groups. Adjusting these variables, make the results comparable between the two strains.

The third paragraph of the Discussion Section, in the previous manuscript, was dedicated to clarify the purpose of these *in vitro* studies, concluding that the *in vivo* results were comparable between both strains.

Table 2- Needs to label Lung and blood as 'Log10 CFU/g' not just 'CFU/g'

Following the Reviewer request, we have modified it.

May 2, 2022

Re: Life Science Alliance manuscript #LSA-2021-01349-TR

Dr. María Eugenia Pachon Ibañez
Institute of Biomedicine of Seville
Infectious Diseases
Avda Manuel Siurot s/n
Seville, Seville 41013
Spain

Dear Dr. Pachon Ibañez,

Thank you for submitting your revised manuscript entitled "IgM-enriched immunoglobulin improves colistin efficacy in pneumonia model by *Pseudomonas aeruginosa*" to Life Science Alliance. The manuscript has been seen by the original reviewers whose comments are appended below. While the reviewers continue to be overall positive about the work in terms of its suitability for Life Science Alliance, some important issues remain.

Our general policy is that papers are considered through only one revision cycle; however, given that the suggested changes are relatively minor, we are open to one additional short round of revision. Please note that I will expect to make a final decision without additional reviewer input upon resubmission.

Please submit the final revision within one month, along with a letter that includes a point by point response to the remaining reviewer comments.

To upload the revised version of your manuscript, please log in to your account: <https://lsa.msubmit.net/cgi-bin/main.plex>
You will be guided to complete the submission of your revised manuscript and to fill in all necessary information.

B. MANUSCRIPT ORGANIZATION AND FORMATTING:

Sincerely,

Reviewer #1 (Comments to the Authors (Required)):

Major comments:

1. The flow diagram for the experimental study has been added as a supplementary figure. However, the authors failed to provide all necessary information according to ARRIVE 2.0 guidelines (<https://dx.plos.org/10.1371/journal.pbio.3000411>) and the experimental design assistant paper (<https://doi.org/10.1371/journal.pbio.2003779>). For instance, the authors do not describe all groups that resulted from the randomization. Please review these papers again and provide a diagram which is clear and complies with all required elements.
2. It could be better to have the flow diagram in the main manuscript since it would help readers understand your experimental design.
3. Dunnett test only provides unidirectional comparisons against a control or standard group. Nonetheless, the Games-Howell test should have been applied for multiple comparisons.
4. The authors did not adequately respond to this prior comment (only a sentence with no additional information was added): "The discussion section of the paper is also lacking the translational viewpoint. Please mention what potential mechanisms may underly your findings since this discussion could be very important to shape the direction of future studies."
5. The authors did not adequately recognize limitations of their study according to ARRIVE 2.0 guidelines which mention: "Item 17b. Comment on the study limitations, including potential sources of bias, limitations of the animal model, and imprecision associated with the results".
6. Generalizability/Translation of the findings is inadequately discussed in the manuscript. Please review the ARRIVE 2.0 Explanation and elaboration document (item 18) to adequately describe this.
7. Please delete the sentence in lines 177-179. Your preclinical model in mice is not comparable to studies in humans. It is important to be aware of the many differences that studies in animals have with respect to those in humans. Furthermore, your study is experimental and the study in humans you are citing is observational, further making them not comparable.
8. There are several imprecisions regarding what the authors say and what the references actually mention. I had already highlighted this problem with several references in the prior version of this manuscript. Please review again all references and make sure that you are adequately using all references, since it is unethical to try to justify your ideas with something that is not described in the work you are citing.
9. For instance, in line 192 the authors cite the congress abstract from Pachon-Ibañez 2014 to say that colistin is a suboptimal antibiotic for severe infections (which is completely untrue). This congress abstract does not mention colistin at all. Please remove this whole phrase as well as the reference.
10. There is an unacceptable number of self-citations (12 in total, corresponding to 33% of references cited in this manuscript). This is unacceptable and reflects that the authors did not perform a proper search of the literature to describe the background of their work and discuss their findings in a fair way. Unjustified self-citations are considered scientific misconduct which violates publication ethics according to COPE.
11. Please read the following series of papers which explain why performing a systematic search of the literature is important to adequately and fairly discuss any research study: DOI 10.1016/j.jclinepi.2020.07.020; DOI 10.1016/j.jclinepi.2020.07.019; DOI 10.1016/j.jclinepi.2020.07.021
12. Since I have significant concerns about scientific misconduct due to excessive self-citations, the authors should perform a reproducible systematic search of studies and describe it according to PRISMA-S (<https://systematicreviewsjournal.biomedcentral.com/articles/10.1186/s13643-020-01542-z>) to show that important literature was not left out and that all self-citations for this research topic are indeed retrieved by performing a systematic search of studies with clear defined criteria in a reproducible way. This search needs to be described in full according to PRISMA-S elements that apply so that it can be reproduced.

Minor comments:

1. Several figures contain bar graphs, however, the authors intended to represent data with their mean and standard error. Thus, there is no justification for using bars alongside the mean and standard error since bar graphs are used to present frequencies or percentages of accumulating data. Since this is not the case, these should be changed for a representation of the mean with its standard error bar only. Please change all figures where this applies.
2. Table 2 does not mention in the legend what test was applied to compare frequencies in the last two columns.
3. Line 41: Please review this phrase since it does not make sense: "with chronic underlying diseases and healthcare-associated infections, including 42 pneumonia, urinary tract infections, and bloodstream infections (BSIs)"
4. Line 66: It should say "have shown".
5. Line 84: Mentioning that something "increased survival of patients" suggests that a causal relationship was shown. This was not shown in the study you are citing since even the authors mention that randomized trials are required. Please correct this.

Reviewer #3 (Comments to the Authors (Required)):

In this paper by Cebrero-Cangueiro et. al. they describe the use of IgM enriched immunoglobulin alone and in concert with two antibiotics as a treatment in a *P. aeruginosa* pneumonia model. As mentioned in the first review the research findings are of interest and importance. The authors have dealt with most of my comments appropriately and Figure 4 and 5 make the data much easier to interpret. A few further comments below

- 1) I still think discussion of the IgM-adjuvant mechanism is lacking. The addition of a couple of sentences in the introduction doesn't really cover this important point. The authors via the sentence in the introduction suggest its through increased phagocytosis and toxin neutralization, but this needs to be discussed after your results as well. The IgM alone did have some

effect in the lung but not blood- does this suggest its helping with phagocytosis in alveolar macrophages for instance?
2) Fig 4 and 5 are a big improvement- however the text under each graph should be in the figure legend.

Reviewer #1 (Comments to the Authors (Required)):**Major comments:**

1. The flow diagram for the experimental study has been added as a supplementary figure. However, the authors failed to provide all necessary information according to ARRIVE 2.0 guidelines (<https://dx.plos.org/10.1371/journal.pbio.3000411>) and the experimental design assistant paper (<https://doi.org/10.1371/journal.pbio.2003779>). For instance, the authors do not describe all groups that resulted from the randomization. Please review these papers again and provide a diagram which is clear and complies with all required elements.

Following the request of the Reviewer, we have modified the diagram according to ARRIVE 2.0 guidelines.

2. It could be better to have the flow diagram in the main manuscript since it would help readers understand your experimental design.

Following the Reviewer request, we have done it (Figure 1 of the current revised manuscript).

3. Dunnett test only provides unidirectional comparisons against a control or standard group. Nonetheless, the Games-Howell test should have been applied for multiple comparisons.

As requested by the Reviewer we have used the Games-Howell test instead of the T3 Dunnett test when comparing the quantitative blood cultures, because the non-homogeneity of variance between groups, as stated in the previous revision of the manuscript.

4. The authors did not adequately respond to this prior comment (only a sentence with no additional information was added): "The discussion section of the paper is also lacking the translational viewpoint. Please mention what potential mechanisms may underly your findings since this discussion could be very important to shape the direction of future studies."

In the previous revision of the manuscript, in response of the comment 2 of the Reviewer #3, we included in the Introduction Section the following sentences about the potential mechanisms underlying the hypothesis of the usefulness of combining antibiotics with IgM-IgG: "IgG is the most abundant antibody and provides protection against infections, bacterial and/or viral. Among its most important functions are to neutralize and eliminate the pathogen, in this case *P. aeruginosa*, toxins generated by the bacteria, and/or other substances produced in inflammatory processes or cell destruction (Birdsall and Casadevall. 2020). Nevertheless, IgG can take some time to form after infection, so combining it with IgM, first line of defense against infection by inhibiting bacterial adhesion to epithelial cells and neutralizing toxins could increase the immune response and be an adjuvant to antibiotics (Birdsall and Casadevall. 2020). Additionally, IgM combined with IgG and IgA have shown opsonophagocytic activity against carbapenem-resistant *P. aeruginosa* (Rossman et al. 2015). These reasons suggest that

using IgM-IG could modulate the immune system of the mice increasing the efficacy of the tested antibiotics.”

In the current revision of the manuscript, following the request of the Reviewer, we have added the following sentences in the Discussion Section: “In these results, we found that the intravenous treatment of a single dose of IgM-IG was able to reduce the bacterial load in lungs against the colistin-susceptible strain Pa147 ($-1.29 \log_{10}$ CFU/g) when compared with the control group. IgM is the first line of defense against infection by inhibiting bacterial adhesion to epithelial cells and neutralizing toxins (Birdsall and Casadevall. 2020). Moreover, IgM is the main immunoglobulin isotype for complement-mediated killing and complement-dependent phagocytosis of infectious bacteria due to its ability for effective complement activation (Spiegelberg, 1989). Furthermore, as is well known, the role of lipopolysaccharide (LPS) in the *P. aeruginosa* pathogenesis, and that polysaccharides (including LPS) are T-cell-independent antigens, being antibodies induced in response to them mostly of the IgM isotype. In acute *P. aeruginosa* infection, the LPS-phenotype predominates and, as such, might be that treating the infection with antibiotics in combination with IgM-IG could be an option for acute infections, because IgM-IG could awaken the immune response faster, helping to resolve it along with antibiotics. In this regard, a murine pneumonia model in BALB/c mice (Horn et al., 2010) by *P. aeruginosa* shows that the monoclonal IgM KBPA101 with exclusive reactivity with LPS of serotype O11, administered previously to the bacterial inoculation, reduced the bacterial lungs and spleen concentrations at 48 hours. In other pneumonia model in C57BL/6 mice (Secher et al., 2011), the same monoclonal IgM KBPA101 could be detected in the bronchoalveolar lavage fluid of *P. aeruginosa* infected mice but not in uninfected animals, after the intravenous administration and, when it was given four hours after the inoculation, decreased ($1.5 \log_{10}$ CFU) significantly the bacterial lungs concentration. IgM KBPA101 could be detected in the bronchoalveolar lavage fluid of infected mice but not in uninfected animals. Additionally, this monoclonal antibody increased the neutrophils recruitment in the lungs compared with control PBS-treated infected mice, while macrophage/monocyte recruitment did not (Secher et al., 2011). Although the IgM-IG is different to monoclonal IgM KBPA101, these data may explain the better results observed, regarding bacterial concentrations, in lungs compared with blood in the present study.” (lines 150-178).

5. The authors did not adequately recognize limitations of their study according to ARRIVE 2.0 guidelines which mention: "Item 17b. Comment on the study limitations, including potential sources of bias, limitations of the animal model, and imprecision associated with the results ".

Following the Reviewer comment, we have modified the limitations of the study as follows: “Although our results suggest that the combined treatment of colistin to IgM-enriched immunoglobulin could improve lung infections by multidrug-resistant *P. aeruginosa*, we have to point out several limitations to the study. The 3R rules (Hubrecht et al. 2019) prevent us from increasing the number of strains in which to confirm the present results; however, to avoid strain-dependent results we have used two different strains in terms of colistin susceptibility and virulence. Also, the 3R rules recommend to avoid other infection models to test the same hypothesis, as well as the increase of the sample size of the experimental groups, although it has not prevented to show better efficacy of the combination of IgM-IG

with colistin, in the performed experiments. Finally, as in any animal model, a limitation is the general caution to translate the preclinical studies to the clinical setting, although the antibiotics dosages have been chosen according to the pharmacodynamics targets in human beings." (lines 224-235).

6. Generalizability/Translation of the findings is inadequately discussed in the manuscript. Please review the ARRIVE 2.0 Explanation and elaboration document (item 18) to adequately describe this.

Following the Reviewer request we included in the Discussion Section the following paragraph: "..., which points to the possible efficacy of antibiotics combined with IgM-IG to diminish the mortality of multidrug-resistant *P. aeruginosa* pneumonia in human beings, which may be as high as 33.3% in patients with ventilator-associated pneumonia treated with colistimethate sodium (Mogyoródi et al. 2022). To extrapolate these results to pneumonia by other multidrug-resistant GNB deserves further research with experimental pneumonia models by other pathogens." (lines 217-223).

7. Please delete the sentence in lines 177-179. Your preclinical model in mice is not comparable to studies in humans. It is important to be aware of the many differences that studies in animals have with respect to those in humans. Furthermore, your study is experimental and the study in humans you are citing is observational, further making them not comparable.

Following the request of the Reviewer, we have deleted the sentence in lines 177-179. Consequently, to make sense the rest of the paragraph, we have modified it as follows: "The only clinical study evaluating the combination of antibiotics with a single dose of IgM-IG was carried out in patients with sepsis or septic shock by multidrug-resistant GNB infections, in which the group that received antibiotics in combination with IgM-IG had a reduction of the mortality rate compared with the group treated with antibiotics alone (Giamarellos-Bourboulis et al. 2016). The caveats of this clinical study are its retrospective design, with cases receiving IgM-IG at the discretion of attending physicians, and the data were collected from a database of ICU-acquired sepsis from 63 study sites. Although 77 % of patients had ventilator-associated pneumonia, only 21 % were caused by *P. aeruginosa*. In this context, in the experimental *P. aeruginosa* pneumonia model, we found that the administration of IgM-IG combined with colistin was useful in reducing the bacterial lung concentration, ..." (lines 207-217).

8. There are several imprecisions regarding what the authors say and what the references actually mention. I had already highlighted this problem with several references in the prior version of this manuscript. Please review again all references and make sure that you are adequately using all references, since it is unethical to try to justify your ideas with something that is not described in the work you are citing.

Following the comment of the Reviewer we have reviewed carefully and corrected all the imprecisions detected in the references.

9. For instance, in line 192 the authors cite the congress abstract from Pachon-Ibañez 2014 to say that colistin is a suboptimal antibiotic for severe infections (which is completely untrue).

This congress abstract does not mention colistin at all. Please remove this whole phrase as well as the reference.

The Reviewer was right. We have deleted it and the sentence, which has been replaced by what is said in the response to the comment 6 of the Reviewer #1.

10. There is an unacceptable number of self-citations (12 in total, corresponding to 33% of references cited in this manuscript). This is unacceptable and reflects that the authors did not perform a proper search of the literature to describe the background of their work and discuss their findings in a fair way. Unjustified self-citations are considered scientific misconduct which violates publication ethics according to COPE.

We are extremely troubled because of the Reviewer suggestion that we behave unethically because of unjustified self-citations. Absolutely, it was not our purpose. This error was due because, as noted by the Reviewer in his/her comments 8 and 9, there were imprecisions and mistakes in the references included in the text.

In this context, we have carefully checked the references, as stated previously. Thus, in the current revised manuscript we cited four own manuscripts out of 35 references. The reasons of them are: i) **Introduction**: Parra-Millán et al. 2022. The impact of the immune response modification by lysophosphatidylcholine in the efficacy of antibiotic therapy in experimental pneumonia by *Pseudomonas aeruginosa* in C57BL/6, developed in our laboratory; ii) **Material and Methods**: López-Rojas et al. 2013. The *Acinetobacter baumannii* CR17, reference strain developed in our laboratory; iii) **Material and Methods**: Gil-Marqués et al. 2018. Model of *Pseudomonas aeruginosa* murine pneumonia, by intra-tracheal instillation in C57BL/6 mice, developed in our laboratory; and iv) **Material and Methods and Discussion**: Pachón-Ibáñez et al. 2018. Colistin pharmacodynamics used for its dosage from one previous study in C57BL/6 mice, developed in our laboratory.

11. Please read the following series of papers which explain why performing a systematic search of the literature is important to adequately and fairly discuss any research study: DOI 10.1016/j.jclinepi.2020.07.020; DOI 10.1016/j.jclinepi.2020.07.019; DOI 10.1016/j.jclinepi.2020.07.021

Please, see the response to the following comment 12.

12. Since I have significant concerns about scientific misconduct due to excessive self-citations, the authors should perform a reproducible systematic search of studies and describe it according to PRISMA-S (<https://systematicreviewsjournal.biomedcentral.com/articles/10.1186/s13643-020-01542-z>) to show that important literature was not left out and that all self-citations for this research topic are indeed retrieved by performing a systematic search of studies with clear defined criteria in a reproducible way. This search needs to be described in full according to PRISMA-S elements that apply so that it can be reproduced.

Following the request of the Reviewer, we have performed a reproducible search of studies according to PRISMA-S. We summarize in following Table the search using the PRISMA-S checklist.

Minor comments:

1. Several figures contain bar graphs, however, the authors intended to represent data with their mean and standard error. Thus, there is no justification for using bars alongside the mean and standard error since bar graphs are used to present frequencies or percentages of accumulating data. Since this is not the case, these should be changed for a representation of the mean with its standard error bar only. Please change all figures where this applies.

Following the request of the Reviewer, we have modified the figures.

2. Table 2 does not mention in the legend what test was applied to compare frequencies in the last two columns.

As requested by the Reviewer we have added it.

3. Line 41: Please review this phrase since it does not make sense: "with chronic underlying diseases and healthcare-associated infections, including 42 pneumonia, urinary tract infections, and bloodstream infections (BSIs)"

As suggested by the Reviewer, to clarify the sentence we have modified the first sentence of this paragraph to "Pseudomonas aeruginosa is a Gram-negative bacillus (GNB) that causes community- and healthcare-associated infections, especially in intensive care units (ICUs) (Bassetti et al. 2016) and in patients with chronic underlying diseases, including pneumonia, urinary tract infections, and bloodstream infections (BSIs) (Mehrad et al. 2015)."

4. Line 66: It should say "have shown".

We have changed it, as requested by the Reviewer.

5. Line 84: Mentioning that something "increased survival of patients" suggests that a causal relationship was shown. This was not shown in the study you are citing since even the authors mention that randomized trials are required. Please correct this.

We have corrected it as suggested by the Reviewer.

Reviewer #3 (Comments to the Authors (Required)):

In this paper by Cebrero-Cangueiro et. al. they describe the use of IgM enriched immunoglobulin alone and in concert with two antibiotics as a treatment in a *P. aeruginosa* pneumonia model. As mentioned in the first review the research findings are of interest and importance. The authors have dealt with most of my comments appropriately and Figure 4 and 5 make the data much easier to interpret. A few further comments below

- 1) I still think discussion of the IgM-adjuvant mechanism is lacking. The addition of a couple of sentences in the introduction doesn't really cover this important point. The authors via the sentence in the introduction suggest its through increased phagocytosis and toxin neutralization, but this needs to be discussed after your results as well. The IgM alone did have some effect in the lung but not blood- does this suggest its helping with phagocytosis in alveolar macrophages for instance?**

Following the request of the Reviewer, to address this important point, we have added the following sentences in the Discussion Section, as also stated in the response to the comment 4 of the Reviewer #1: "In these results, we found that the intravenous treatment of a single dose of IgM-IG was able to reduce the bacterial load in lungs against the colistin-susceptible strain Pa147 ($-1.29 \log_{10}$ CFU/g) when compared with the control group. IgM is the first line of defense against infection by inhibiting bacterial adhesion to epithelial cells and neutralizing toxins (Birdsall and Casadevall. 2020). Moreover, IgM is the main immunoglobulin isotype for complement-mediated killing and complement-dependent phagocytosis of infectious bacteria due to its ability for effective complement activation (Spiegelberg, 1989). Furthermore, as is well known, the role of lipopolysaccharide (LPS) in the *P. aeruginosa* pathogenesis, and that polysaccharides (including LPS) are T-cell-independent antigens, being antibodies induced in response to them mostly of the IgM isotype. In acute *P. aeruginosa* infection, the LPS-phenotype predominates and, as such, might be that treating the infection with antibiotics in combination with IgM-IG could be an option for acute infections, because IgM-IG could awaken the immune response faster, helping to resolve it along with antibiotics. In this regard, a murine pneumonia model in BALB/c mice (Horn et al., 2010) by *P. aeruginosa* shows that the monoclonal IgM KBPA101 with exclusive reactivity with LPS of serotype O11, administered previously to the bacterial inoculation, reduced the bacterial lungs and spleen concentrations at 48 hours. In other pneumonia model in C57BL/6 mice (Secher et al., 2011), the same monoclonal IgM KBPA101 could be detected in the bronchoalveolar lavage fluid of *P. aeruginosa* infected mice but not in uninfected animals, after the intravenous administration and, when it was given four hours after the inoculation, decreased ($1.5 \log_{10}$ CFU) significantly the bacterial lungs concentration. IgM KBPA101 could be detected in the bronchoalveolar lavage fluid of infected mice but not in uninfected animals. Additionally, this monoclonal antibody increased the neutrophils recruitment in the lungs compared with control PBS-treated infected mice, while macrophage/monocyte recruitment did not (Secher et al., 2011). Although the IgM-IG is different to monoclonal IgM KBPA101, these data may explain the better results observed, regarding bacterial concentrations, in lungs compared with blood in the present study." (lines 150-178).

2) Fig 4 and 5 are a big improvement- however the text under each graph should be in the figure legend.

As requested by the Reviewer, we have done it.

June 1, 2022

RE: Life Science Alliance Manuscript #LSA-2021-01349-TRR

Dr. María Eugenia Pachon Ibañez
Institute of Biomedicine of Seville
Infectious Diseases
Avda Manuel Siurot s/n
Seville, Seville 41013
Spain

Dear Dr. Pachon Ibañez,

Thank you for submitting your revised manuscript entitled "IgM-enriched immunoglobulin improves colistin efficacy in pneumonia model by *Pseudomonas aeruginosa*". We would be happy to publish your paper in Life Science Alliance pending final revisions necessary to meet our formatting guidelines.

- please use the [10 author names, et al.] format in your references (i.e. limit the author names to the first 10)
- please consult our manuscript preparation guidelines <https://www.life-science-alliance.org/manuscript-prep> and make sure your manuscript sections are in the correct order
- please add your figure (main and supplementary) and table legends to the main manuscript text
- please provide an approval statement for the animal experiments

A. FINAL FILES:

B. MANUSCRIPT ORGANIZATION AND FORMATTING:

Sincerely,

June 7, 2022

RE: Life Science Alliance Manuscript #LSA-2021-01349-TRRR

Dr. María Eugenia Pachon Ibañez
Institute of Biomedicine of Seville
Infectious Diseases
Avda Manuel Siurot s/n
Seville, Seville 41013
Spain

Dear Dr. Pachon Ibañez,

Thank you for submitting your Research Article entitled "IgM-enriched immunoglobulin improves colistin efficacy in pneumonia model by *Pseudomonas aeruginosa*". It is a pleasure to let you know that your manuscript is now accepted for publication in Life Science Alliance. Congratulations on this interesting work.

DISTRIBUTION OF MATERIALS:

Again, congratulations on a very nice paper. I hope you found the review process to be constructive and are pleased with how the manuscript was handled editorially. We look forward to future exciting submissions from your lab.

Sincerely,
